# Altered Expression of the *MEG3*, *FTO*, *ATF4*, and Lipogenic Genes in PBMCs from Children with Obesity and Its Associations with Added Sugar Intake

**DOI:** 10.3390/nu17152546

**Published:** 2025-08-02

**Authors:** Adrián Hernández-DíazCouder, Pablo J. Paz-González, Maryori Valdez-Garcia, Claudia I. Ramírez-Silva, Karol Iliana Avila-Soto, Araceli Pérez-Bautista, Miguel Vazquez-Moreno, Ana Nava-Cabrera, Rodrigo Romero-Nava, Fengyang Huang, Miguel Cruz

**Affiliations:** 1Unidad de Investigación Médica en Bioquímica, Hospital de Especialidades, Centro Médico Nacional Siglo XXI, Instituto Mexicano del Seguro Social, Mexico City 06720, Mexico; adrian.hernandez.diazc@gmail.com (A.H.-D.);; 2Instituto Nacional de Salud Pública, Cuernavaca 62100, Mexico; 3OOAD DF Norte del Instituto Mexicano del Seguro Social, Unidad de Medicina Familiar No. 23., Mexico City 07070, Mexico; 4Laboratorio de Investigación en Genética de Enfermedades Metabólicas, Sección de Estudios de Posgrado e Investigación, Escuela Superior de Medicina, Instituto Politécnico Nacional, Mexico City 11340, Mexico; 5Laboratorio de Investigación en Obesidad y Asma, Hospital Infantil de México Federico Gómez, Mexico City 06720, Mexico

**Keywords:** gene regulation, pediatric population, non-coding RNA, dietary sugar consumption

## Abstract

Background: Obesity and its complications have increased in both adults and children, with pediatric populations developing metabolic disorders at earlier ages. Long non-coding RNAs, particularly *MEG3*, are involved in obesity through regulation of lipogenic genes including *ATF4*, *FTO*, *SREBP1*, *FASN*, and *ACACA*. However, data on *MEG3* expression in pediatric obesity are limited. This study evaluated *MEG3*, *FTO*, and *ATF4* expression in PBMCs from children with obesity and their associations with added sugar intake and lipid metabolism genes. Methods: In this cross-sectional study 71 children within the age range of 6 to 12 years were included (28 normal weight and 43 with obesity). Anthropometrical and clinical parameters and dietary added sugar consumption were analyzed. Real-time PCR was performed to assess *MEG3*, *FTO*, *ATF4*, *SREBP1*, *FASN*, and *ACACA* gene expression in peripheral blood mononuclear cells. Results: The expression of *MEG3*, *ATF4*, *FTO*, *SREBP1*, *FASN*, and *ACACA* was decreased in children with obesity. *MEG3* and *FTO* showed sex-dependent expression in children without obesity, while additional sex-related differences were observed for *SREBP1*, *FASN*, *ACACA*, *FTO*, and *MEG3* in children with obesity. *MEG3* was associated with the expression of *SREBP1*, *FASN*, *ACACA*, *FTO*, and *ATF4*. In insulin-resistant (IR) children, *MEG3*, *ATF4*, *FTO*, *ACACA*, and *SREBP1* were reduced, while *FASN* was increased. Added sugar intake negatively correlated with *FTO*, *SREBP1*, and *ACACA*. Conclusions: The *MEG3*, *FTO*, and *ATF4* expression was altered in children with obesity, showing sex- and IR-related differences. Added sugar intake correlated negatively with lipogenic gene expression.

## 1. Introduction

Obesity and its related diseases have increased in recent decades, in both adults and children, becoming a major public health concern. Epidemiological studies have demonstrated a considerable global increase in the number of children affected by overweight and obesity. For instance, the prevalence of obesity in children aged 5–14 years has tripled worldwide, increasing from 2.0% in 1990 to 6.8% in 2021, and it is estimated that, by 2050, the prevalence will reach 15.6% [1]. Furthermore, 18.1% of children between 5 and 11 years of age have been reported to have obesity [2]. Among the primary risk factors, obesity is strongly associated with the onset of insulin resistance (IR), type 2 diabetes (T2D), dyslipidemia, and cardiovascular disease. The pediatric population in Mexico is prone to developing these metabolic complications at earlier ages [3]. As for genetic factors, several genes involved in lipid metabolism have been identified as potential contributors to the development of obesity and its metabolic complications [4].

Long non-coding RNAs (lncRNAs), which are RNA molecules longer than 200 nucleotides, do not encode proteins but play important regulatory roles at the transcriptional, post-transcriptional, and epigenetic levels [5]. They can act as molecular scaffolds, guides, decoys, or signals, influencing gene expression, chromatin remodeling, and RNA splicing [6]. LncRNAs are involved in several biological processes, such as the regulation of lipid and glucose metabolism and inflammatory pathways—hallmarks of IR and obesity [7].

Maternally expressed gene 3 (*MEG3*), a prominent gene located on chromosome 14 in humans, is subject to genomic imprinting and is expressed only from the maternal allele, while the paternal allele is silenced by methylation [8]. *MEG3* transcripts are approximately 700 nucleotides in length and are highly conserved across species [8]. *MEG3* is expressed in several tissues, including peripheral blood mononuclear cells (PBMCs) [9,10,11], and has been implicated in the pathogenesis of obesity and obesity-related diseases, including cardiovascular diseases and T2D [12,13,14], and its differential expression offers significant potential for its use as both a biomarker and a therapeutic target [15]. *MEG3* expression has primarily been studied in adult populations and there is limited evidence in pediatric populations in this context.

In addition to genetic factors, dietary factors also play a key role in the development of obesity and its metabolic complications. High intake of added sugars has been associated with an increased risk of obesity, IR, and dyslipidemia, especially in children [16,17]. Emerging evidence suggests that excessive consumption of added sugars may alter lipid metabolism and contribute to the dysregulation of gene expression, including lncRNAs, such as *MEG3* [18].

In addition, biological factors such as age and sex influence gene expression and metabolic responses. During puberty, hormonal changes—such as increased estrogen and testosterone—can modify fat distribution [19], insulin sensitivity [20,21], and lipid levels [20], contributing to sex- and age-specific metabolic profiles. Several studies have reported that the expression of metabolic genes, including *FTO* [22] and *SREBP1c* [23], varies by sex.

Recent studies have demonstrated that *MEG3* modulates key genes involved in lipogenesis, including *ATF4*, *FTO*, *SREBP1c*, *FASN*, and *ACACA*, through mechanisms involving microRNA interaction and epigenetic regulation. For instance, *ATF4* is a transcription factor involved in cellular stress responses. Although not a classical lipid metabolism gene, it contributes to lipogenesis [24]. Its deficiency has been associated with reduced hepatic lipogenesis through downregulation of *SREBP1c*, *PPARγ*, and *FASN* expression [25]. Meanwhile, *FTO*, a gene identified through genome-wide association studies (GWASs) as associated with obesity, plays a role in lipid accumulation by enhancing the maturation and nuclear translocation of *SREBP1c* [26]. Unlike causal genes for monogenic obesity, such as *MC4R* or *LEP*, which have a direct role in early-onset and severe forms of obesity, *FTO* influences susceptibility through regulatory mechanisms [27]. These findings suggest a potential regulatory pathway involving *MEG3*, *FTO*, and *ATF4*, in which *MEG3* may modulate lipid-metabolism-related genes through epigenetic or post-transcriptional mechanisms. Given these findings, the evaluation of the expression of these genes in PBMCs allows for a minimally invasive approach to explore molecular alterations related to obesity and dietary factors in children. Accordingly, the present study aimed to evaluate the expression of *MEG3*, *FTO*, and *ATF4* in PBMCs from children with obesity and to assess their associations with added sugar intake and the expression of lipogenesis-related genes.

## 2. Materials and Methods

### 2.1. Study Participants

The study population comprised 71 non-related children between the ages of 6 and 12 enrolled from Mexico City (28 without obesity and 43 with obesity) to evaluate the association between the consumption of added sugar and gene expression related to lipogenesis, obesity, and cardiometabolic risk factors. The study received ethical approval from the Ethics Committee of the Instituto Mexicano del Seguro Social (R-2024-785-061) and was conducted in compliance with the Declaration of Helsinki. Assent was obtained from all participating children, and informed consent was obtained from their parents or legal guardians. Children presenting with acute infections, chronic diseases, participation in a weight loss program, or fasting plasma glucose levels exceeding 126 mg/dL were excluded from the study.

### 2.2. Anthropometric Measurements

Anthropometric measurements were conducted following a standardized protocol to ensure accuracy, consistency, and data reliability. The procedure was carried out by trained personnel with expertise in anthropometric techniques, using calibrated instruments under controlled conditions.

Each child participant underwent measurement of body weight, height, waist circumference, and hip circumference in a single session at the beginning of the study. Body weight was measured using a digital scale with a precision of 100 g (Seca, Hamburg, Germany). Children were weighed wearing light clothing and no shoes, standing upright in the center of the scale platform, maintaining a straight posture without movement during the reading. For height measurement, a portable device with 0.1 cm resolution was utilized (Seca 225, Hamburg, Germany). The child stood barefoot with heels together, back straight, and head aligned in the horizontal plane (Frankfurt plane), ensuring contact of the heels, buttocks, scapulae, and head with the vertical surface. Waist circumference was measured using a 0.1 cm precision flexible tape measure, resistant to stretching, placed around the abdomen just above the upper edge of the iliac crest, and recorded at the end of a gentle expiration. The child stood upright in a relaxed posture, with arms hanging loosely at the sides and feet slightly apart. Hip circumference was measured with the same type of tape, placed horizontally around the most prominent part of the buttocks, ensuring the tape was level and did not compress the soft tissue. As with other measurements, the child stood still and relaxed throughout the procedure. Each measurement was recorded immediately on an individual data collection sheet and double-checked to ensure accuracy. Body mass index (BMI) was calculated as weight (kg) divided by height squared (m^2^). BMI values were converted into age- and sex-specific percentiles. The data were then processed using the WHO Anthro software (version 3.2) and evaluated according to the World Health Organization (WHO) growth standards [28]. Children were classified as having normal weight (BMI between the 5th and 85th percentiles; BMI category 1) or obesity (BMI ≥ 97th percentile; BMI category 3) following the WHO criteria for overweight and obesity in children aged 5 to 19 years [28].

### 2.3. Biochemical Measurements

Venous blood samples were collected in the morning following an 8 h fast to determine levels of low-density lipoprotein cholesterol (LDL-C), high-density lipoprotein cholesterol (HDL-C), total cholesterol (Total CHO), triglycerides (TG), glucose, and insulin. Serum concentrations of LDL-C, HDL-C, Total CHO, TG, and glucose were determined using an enzymatic colorimetric method on the ILab 350 Clinical Chemistry System (Instrumentation Laboratory IL, Barcelona, Spain). Fasting insulin levels were measured using the Elecsys Insulin Assay on the Cobas e411 immunoassay analyzer (Roche Diagnostics GmbH, Mannheim, Germany). To determine the HOMA-IR, fasting glucose concentration (mg/dL) was multiplied by the fasting insulin concentration (µU/mL) and divided by 405. IR was also evaluated in children, according to HOMA-IR values: those with HOMA-IR ≤ 3 were considered not to have IR, and those with HOMA-IR > 3 were considered to have IR.

### 2.4. Dietary Added Sugar Consumption

Dietary added sugar consumption was assessed using a semi-quantitative food frequency questionnaire (FFQ) that inquired about food intake over the past month [29]. It was designed using the Food Frequency Questionnaire from the Mexican National Health and Nutrition Survey (ENSANUT) for school-aged children (from the National Institute of Public Health) as a reference. Reference portions were estimated based on 24 h dietary recalls from ENSANUT for school-aged children, and all portions were expressed in household measurements.

To estimate added sugar intake, a specific food composition database was developed to process the data collected through the questionnaire. This database was built using the Mexican Food Composition Table (BAM) [30]. The FFQ was analyzed following the same methodological criteria used in the analysis of grams and nutrients in the ENSANUT FFQ [31].

### 2.5. RT-qPCR

RNA extraction from whole venous blood was performed using the TRIzol reagent (Thermo Scientific, Waltham, MA, USA), according to the instructions provided by the manufacturer. RNA was extracted and its quality was analyzed with a NanoDrop 1000 spectrophotometer (Thermo Scientific, Waltham, MA, USA). Genomic DNA was removed from RNA samples by treatment with DNase I (Thermo Fisher Scientific, Waltham, MA, USA), following the manufacturer’s protocol. The integrity of the RNA was confirmed by agarose gel electrophoresis. Subsequently, 500 ng of total RNA were reverse transcribed into cDNA using the RevertAid First Strand cDNA Synthesis Kit (Thermo Scientific, Waltham, MA, USA). The synthesized cDNA was employed as a template for RT-qPCR using the Maxima SYBR Green/ROX qPCR Master Mix (Thermo Scientific, Waltham, MA, USA) on a 7900HT Fast Real-Time PCR System (Applied Biosystems, Foster City, CA, USA). The following primers were employed to assess gene expression: *FTO* (Assay ID: QT00088802), *ATF4* (QT00074466), *SREBP1* (QT00036897), *FASN* (QT00030618), *ACACA* (QT01670053), *RPLP0* (QT01839887), and *MEG3* (LPH02974A), all obtained from Qiagen (Hilden, Germany). mRNA relative concentrations were normalized with Ct values of *RPLP0*, and values were calculated using the 2^−ΔΔCt^ formula.

### 2.6. Statistical Analysis

The normality of continuous variables was assessed using the Shapiro–Wilk test. Variables with normal distribution were analyzed using the Student’s *t*-test, whereas variables that did not follow a normal distribution were examined using the Mann–Whitney U test. The chi-square test was used to assess differences in categorical variables. Pearson’s correlation was applied for variables with a normal distribution, while Spearman’s correlation was used for variables with a non-normal distribution. Differences in gene expression across groups were analyzed through two-way ANOVA (sex and obesity status as factors), with Tukey’s post hoc test applied for multiple comparisons.

Due to the non-normal distribution of residuals in the linear regression model, quantile regression was used to evaluate associations across different points of the outcome distribution, adjusting for age, sex, obesity, and total energy intake. No variable standardization was applied. For all analyses, a two-tailed *p*-value < 0.05 was considered statistically significant. All analyses were conducted using Stata software (version 19.5; StataCorp LLC, College Station, TX, USA).

## 3. Results

### 3.1. Description of Study Subjects

The health outcome variables of the children are shown in Table 1. The mean age was 8.94 ± 1.68 years. Girls represented 60.56% of the sample, while boys accounted for 39.44%. BMI, insulin, HOMA-IR, and TG levels were significantly higher in the obesity group compared to the without obesity group. Moreover, HDL-C concentration was significantly lower in children with obesity than in those without obesity. No differences were found in glucose, total cholesterol, or LDL-C between the two groups. Finally, no significant differences in added sugar and total energy intake were observed between the groups.

### 3.2. Expression of Genes Involved in Lipid Metabolism in Children with Obesity

Results showed alterations in gene expression associated with lipid metabolism in children with obesity. As shown in Figure 1, the mRNA expression of *SREBP1* (*p* = 0.0176), *FASN* (*p* = 0.0412), *ACACA* (*p* = 0.0255), *FTO* (*p* < 0.0001), and *ATF4* (*p* < 0.0001) was significantly reduced in the obesity group compared to the control group. Moreover, the expression of the lncRNA *MEG3* (*p* = 0.0197) was found to be higher among children with obesity than among those without obesity.

In addition, gene expression was also analyzed according to sex in both the normal weight and obesity groups. As shown in Figure 2, boys with obesity exhibited lower *SREBP1* expression compared to boys without obesity (*p* = 0.015), girls with obesity (*p* = 0.018), and girls without obesity (*p* = 0.023). For *FASN*, boys with obesity showed higher expression than girls with obesity (*p* = 0.005), while girls with obesity had lower levels than girls without obesity (*p* = 0.017). *ACACA* expression was also lower in boys with obesity than in boys without obesity (*p* = 0.001), girls with obesity (*p* = 0.011), and girls without obesity (*p* = 0.003). Regarding *FTO*, boys with obesity had lower expression than boys without obesity (*p* < 0.001) and girls without obesity (*p* < 0.001) but higher expression than girls with obesity (*p* = 0.039). Boys without obesity showed higher *FTO* expression than girls with or without obesity (both *p* < 0.001), and girls with obesity had reduced levels compared to girls without obesity (*p* < 0.001). For *ATF4*, boys with obesity showed lower expression than boys without obesity (*p* = 0.036) and girls without obesity (*p* < 0.001). Girls with obesity also had lower expression than both girls without obesity (*p* < 0.001) and boys without obesity (*p* = 0.005). Finally, *MEG3* expression was higher in boys with obesity compared to boys without obesity (*p* = 0.002) and girls with obesity (*p* = 0.027), while girls with obesity showed higher *MEG3* levels than boys without obesity (*p* = 0.014).

Using the IR cutoff, 50.70% of participants were identified as having IR. As shown in Figure 3, the expression of *SREBP1* (*p* = 0.028), *ACACA* (*p* = 0.038), *FTO* (*p* = 0.045), *ATF4* (*p* = 0.041), and *MEG3* (*p* = 0.021) was reduced in children with IR compared to those without IR.

### 3.3. Associations Between lncRNA MEG3 and Gene Expression and Biochemical Parameters

The quantile regression analysis adjusted for sex, age, and obesity between the expression of lncRNA *MEG3* and of genes involved in lipogenesis was examined and the findings are shown in Table 2. There was a positive association with the expression of *SREBP1* (β = 0.322; *p* < 0.001), *FASN* (β = 0.178; *p* = 0.002), *ACACA* (β = 0.180; *p* = 0.030), *FTO* (β = 0.146; *p* = 0.004), and *ATF4* (β = 0.149; *p* = 0.001). Exploratory Spearman’s correlations between lncRNA *MEG3* expression and lipogenesis-related genes are presented in Appendix A. In these analyses, *MEG3* expression showed positive associations with *SREBP1*, *FASN*, *FTO*, and *ATF4*.

Regarding the relationship between lipogenesis-related genes and clinical and metabolic parameters, quantile regression adjusted for sex, age, and obesity revealed that *SREBP1* was negatively associated with glucose (β = −0.020; *p* = 0.026), insulin (β = −0.012; *p* = 0.030), and HOMA-IR (β = −0.050; *p* = 0.049). *FASN* was negatively associated with HDL-C (β = −0.008; *p* = 0.037). No significant associations were observed for *ACACA*, *FTO*, *ATF4*, or *MEG3* (*p* > 0.05) as shown in Table 3. Exploratory Spearman’s correlations between lncRNA *MEG3*, lipogenesis-related genes, and clinical parameters are shown in Appendix A. *SREBP1* correlated inversely with glucose, insulin, and HOMA-IR, while *FASN* correlated positively with glucose and HOMA-IR. *ACACA* showed inverse correlations with anthropometric and insulin resistance markers and a positive correlation with HDL-C. *FTO* and *ATF4* were negatively correlated with adiposity and insulin resistance indices. *MEG3* showed no significant correlations with clinical variables.

### 3.4. Association Between Added Sugar Intake and Molecular and Biochemical Parameters

Although no significant differences in added sugar intake were observed between children without obesity and those with obesity (Table 1), quantile regression analysis adjusted for sex and total energy intake revealed significant associations with gene expression, as shown in Table 4. There was a significant negative association between added sugar intake and *SREBP1* (β = −0.360; *p* = 0.050) and *FTO* (β = −0.221; *p* = 0.032), whereas no significant associations were observed for *FASN*, *ACACA*, *ATF4*, and *MEG3*. Exploratory correlations between added sugar intake and gene expression are presented in Appendix A. Added sugar intake was inversely correlated with *SREBP1*, *ACACA*, and *FTO*, while no significant correlations were observed for *FASN*, *ATF4*, or *MEG3*.

## 4. Discussion

Emerging studies in both animal models and humans have begun to elucidate the molecular mechanisms of the lncRNA *MEG3* in obesity and obesity-related diseases [32]. However, its clinical relevance in the context of pediatric obesity remains poorly understood.

In this study, the expression of *MEG3*, known to be involved in lipogenesis, was evaluated in PBMCs from children with obesity and compared with that of children without obesity. Our results show an elevated expression of *MEG3* in children with obesity, which is consistent with recent findings reporting elevated expression of *MEG3* in the serum of children with obesity [9]. Interestingly, we observed a sex-dependent expression pattern: in the control group, girls showed higher expression levels than boys, whereas in the obesity group, boys had higher *MEG3* expression than girls. Moreover, boys with obesity showed increased *MEG3* expression compared to boys without obesity. This sex-specific shift in *MEG3* expression from girls in the control group to boys in the obesity group may reflect differential regulatory mechanisms influenced by sex hormones, the onset of puberty, or distinct inflammatory responses in boys and girls with obesity.

Previous studies have reported a positive correlation between *MEG3* and lipogenic genes, such as *FASN* and *PPARG*, in the subcutaneous adipose tissue of females with obesity [33]. In line with these observations, our data showed that *MEG3* expression in PBMCs was associated with the expression of *SREBP1*, *FASN*, *ACACA*, *FTO*, and *ATF4* in children with obesity, but no significant correlations were found with anthropometric or biochemical parameters.

Experimental studies on in vivo and in vitro models have demonstrated that *Meg3* inhibits de novo lipogenesis by reducing the expression of *Acc* and *Fasn* [12,13]. Furthermore, silencing *Meg3* in mice fed a high-fat diet led to reduced body weight, lower glucose and insulin levels, decreased inflammation, and diminished fat accumulation [14].

Our results showed that children with IR exhibited lower *MEG3* expression. This finding contrasts with increased *MEG3* levels in adult patients with T2D [10,11]. In animal studies, *Meg3* upregulation worsens hepatic IR by increasing *Foxo1* expression [12]. Additionally, *Meg3* can act as a competing endogenous RNA, modulating miR-214 and consequently inducing *Atf4* expression, thereby contributing to hepatic IR [34]. Therefore, the evidence suggests that reduced expression of *MEG3* could be related to de novo lipogenesis present in children with obesity and IR. However, more experimental studies in humans are needed to elucidate the function of *MEG3* in this context.

Regarding *ATF4* gene, our results showed decreased expression in PBMCs from children with obesity compared to those without obesity. In contrast, a previous study reported no changes in *ATF4* expression in women with obesity compared to those with normal weight [35]. Also, that study showed a positive correlation between *ATF4* expression and BMI, cholesterol, triglycerides, and HDL-C [35]. In contrast to these observations, our data revealed no significant associations between *ATF4* expression and these lipid parameters.

Additionally, when stratified by sex, girls with obesity showed lower *ATF4* expression compared to girls and boys with normal weight, while boys with obesity exhibited decreased expression compared to boys with normal weight. This sex-dependent expression pattern could suggest distinct regulatory responses that may influence lipid metabolism differently in boys and girls with obesity.

*ATF4* is a transcription factor involved in cellular stress responses but also contributes to the regulation of lipogenesis [24]. Previous experimental studies conducted in *Atf4*-deficient animal models demonstrated a diminished hepatic lipogenesis induced by high fructose intake, due to reduced expression of *Srebp1c*, *Pparγ*, and *Fasn* [25]. In vitro experiments reported that *Atf4* inhibits the transcription of *Srebp1*, a central regulator of fatty acid synthesis [36]. In our study, *ATF4*, *SREBP1*, and *FASN* expression was reduced in PBMCs from children with obesity, contrasting with reports showing no differences in women with obesity [37]. Interestingly, we observed a significant positive correlation between *ATF4* and *SREBP1*, *FASN*, and *ACACA* expression in PBMCs from children with obesity (Appendix A). Moreover, *SREBP1* and *FASN* expression correlated with glucose homeostasis markers, whereas *ACACA* expression correlated with glucose homeostasis markers and clinical and lipid parameters (Appendix A). After adjustment for sex, age, and obesity, *SREBP1* expression showed negative associations with glucose homeostasis markers, whereas *FASN* expression was inversely associated with HDL-C. In contrast, *ATF4* expression did not show significant associations with any variable. These findings suggest that the *ATF4–SREBP1* interaction may be regulated differently in pediatric obesity, possibly reflecting early compensatory mechanisms.

On the other hand, we reported that children with IR presented decreased *ATF4* expression compared to those without IR. Conversely, recent studies have reported increased *ATF4* expression in PBMCs from adult patients with T2D compared to non-diabetic subjects [11,38]. One of those studies also found a positive correlation between *ATF4* expression and *MEG3* expression [11]. Consistently, we also observed a significant correlation between *ATF4* and *MEG3* expression in PBMCs from children [11]. These results suggest that *MEG3* may regulate *ATF4* expression through the molecular mechanisms mentioned above. Interestingly, in contrast to adult data [11], we found negative correlations between *ATF4* expression and insulin levels, HOMA-IR values, and BMI in the pediatric population.

Experimental studies have reported that the inhibition of *Atf4* reversed hepatic IR induced by acute brain endoplasmic reticulum stress [39]. Conversely, a recent study in β-cell-specific *Atf4*-deficient animal models exhibited exacerbated diabetes, evidenced by hyperglycemia, as well as markers of dedifferentiation [40,41]. Furthermore, *Atf4* in β-cells is required for glucose homeostasis during aging and metabolic stress but not in young mice [41]. Interestingly, recent in vitro experiments have also suggested a regulatory interplay between *Fto* and *Atf4*, showing that *Fto* expression may promote gluconeogenesis by inhibiting *Atf4* expression [42]. Therefore, our results show an opposing trend between children and adults, which may reflect developmental or compensatory differences in the regulation of *ATF4*. This discrepancy may suggest that, in early stages of metabolic dysfunction, reduced *ATF4* expression represents an adaptive response aimed at mitigating cellular stress or limiting further metabolic impairment.

Regarding the *FTO* gene, a previous study demonstrated that high *FTO* expression in PBMCs is associated with increased body mass [43]. In contrast, our results showed decreased *FTO* expression levels in PBMCs from children with obesity, with a significant negative correlation observed with BMI and waist and hip circumference. Conversely, studies in adolescent and adult patients with overweight and obesity have reported no significant changes in *FTO* expression in PBMCs and adipose tissue compared to normal weight subjects [44,45,46,47]. These findings suggest that the reduced *FTO* expression observed in children may reflect age-specific regulatory mechanisms of *FTO* in early stages of metabolic dysregulation.

Interestingly, FTO expression exhibited a sex-dependent pattern: boys without obesity showed the highest expression levels, followed by boys with obesity, while girls with obesity had the lowest expression levels. This trend reinforces the potential role of sex-specific regulatory mechanisms in the expression of *FTO* in pediatric obesity.

In addition, experimental research suggests that *Fto* drives lipid accumulation in hepatocytes by supporting the nuclear activation and maturation of *Srebp1c* [26]. Consistent with these findings, our results showed a strong positive correlation between *FTO* and *SREBP1*, *FASN*, and *ACACA* in PBMCs (Appendix A). However, after adjusting for sex, age, and obesity *FTO* expression did not show significant associations with any variable.

On the other hand, recent experimental studies in animal models have demonstrated that m6A modification of *Meg3* by demethylase activity of *Fto* reduces *Meg3* expression [48]. Conversely, our study found a positive correlation between *FTO* and *MEG3* expression in PBMCs from children with obesity. These findings suggest that the regulation of *MEG3* expression by *FTO* may differ in humans. However, further studies are needed to elucidate the role of *FTO* activity in regulating *MEG3* expression in this context.

A previous study reported increased mRNA and protein expression of *FTO* in subcutaneous adipose tissue from women with higher HOMA-IR values. In that study, a positive correlation was also found between *FTO* expression and HOMA-IR index [49]. Conversely, our findings showed reduced *FTO* expression in PBMCs from children with IR compared to those without IR. Additionally, we found a strong negative correlation between *FTO* expression and both insulin levels and HOMA-IR values. In line with these observations, a recent study reported downregulation of *FTO* expression in human pancreatic islets from patients with diabetes [50]. That study also demonstrated that reduced *Fto* expression impaired insulin release in rat pancreatic cells [50]. Therefore, these findings suggest that *FTO* may play an important role in the development of T2D. However, further experimental and clinical studies are needed to elucidate the role in both children and adults in this context.

In addition to these molecular associations, we evaluated the potential influence of dietary added sugar intake on gene expression. Our analysis showed significant negative correlations between added sugar intake and the expression of *SREBP1*, *ACACA*, and *FTO* in PBMCs from children with obesity (Appendix A). Similar findings have been reported in adipose tissue, where a significant negative correlation was observed between *FTO* expression in adipose tissue and total carbohydrate intake in subjects with and without obesity [47]. Furthermore, after adjusting for sex and total energy intake, dietary intake of added sugar was negatively associated with the *FTO* and *SREBP1* expression in PBMCs from children with obesity. Specifically, intake > 50 g of added sugar was associated with reductions of 0.221 and 0.360 units in *FTO* and *SREBP1* expression levels in PBMCs, respectively. These findings suggest that excessive added sugar intake may contribute to early dysregulation of lipid metabolism pathways mediated by FTO and SREBP1 in pediatric obesity. Although our analyses were adjusted for sex and age, the lack of pubertal assessment (e.g., Tanner staging) may limit the interpretation of some findings, particularly those related to metabolic and gene expression variability. Future studies should consider incorporating pubertal status to better characterize developmental influences on gene regulation.

The main limitations of this study are that gene expression was analyzed in PBMCs, which may not fully reflect expression patterns in primary metabolic tissues such as the liver, adipose tissue, or muscle. However, PBMCs are a relevant surrogate for investigating systemic metabolic alterations, particularly in pediatric populations where invasive sampling is not feasible. Because the study is cross-sectional, it does not allow for conclusions about causality between gene expression, metabolic parameters, and added sugar intake. Additionally, dietary intake was assessed using food frequency questionnaires, which may be affected by recall bias and reporting errors, especially in children. Another limitation is that pubertal status was not assessed using Tanner staging. While this could have provided a more precise understanding of biological variability and metabolic changes, its implementation would have required clinical evaluations and handling of sensitive information, which could have reduced participation rates. Instead, analyses were adjusted for age and sex to account for developmental differences. Moreover, this study focused on genes related to lipogenesis and the metabolic stress response but did not include key genes involved in lipoprotein metabolism, such as *LDLR* or *APOB*, which limits the scope of metabolic pathway analysis. Notably, a strength of this study is that it is among the first to examine the association between added sugar intake and the expression of genes related to metabolic disease in a pediatric population, providing insight into early molecular alterations potentially linked to diet.

## 5. Conclusions

Our findings indicate that the expression of *MEG3*, *FTO*, and *ATF4* is altered in children with obesity. Both *MEG3* and *FTO* exhibit a sex-dependent expression pattern in children without obesity, while in children with obesity, this sex-related variation is also observed for *SREBP1*, *FASN*, *ACACA*, *FTO*, and *MEG3*, suggesting a potential role of sex in the regulation of lipogenic genes under obesogenic conditions. In children with IR, *FASN* expression was increased, whereas *SREBP1*, *ATF4*, *ACACA*, *FTO*, and *MEG3* expressions were reduced. Moreover, added sugar intake correlated negatively with *SREBP1* and *FTO*, suggesting a potential modulatory effect of dietary sugar on lipogenic gene expression. Taken together, these findings suggest that dietary sugar intake, gene expression, and sex differences may interact and contribute to early metabolic alterations in children with obesity.

## Figures and Tables

**Figure 1 nutrients-17-02546-f001:**
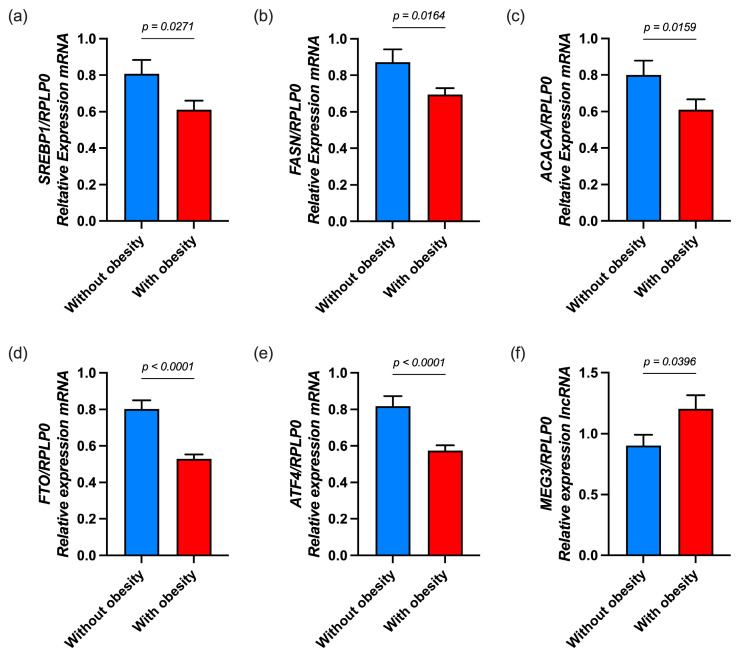
The expression of lipogenic genes in children with and without obesity. Gene expression of (**a**) *SREBP1*, (**b**) *FASN*, (**c**) *ACACA*, (**d**) *FTO*, (**e**) *ATF4*, and (**f**) *MEG3.* mRNA expression was measured by RT-qPCR using *RPLP0* as the reference gene and calculated using the 2^−ΔΔCt^ method. Statistical differences were analyzed using the Mann–Whitney U test. Results are presented as mean ± SD.

**Figure 2 nutrients-17-02546-f002:**
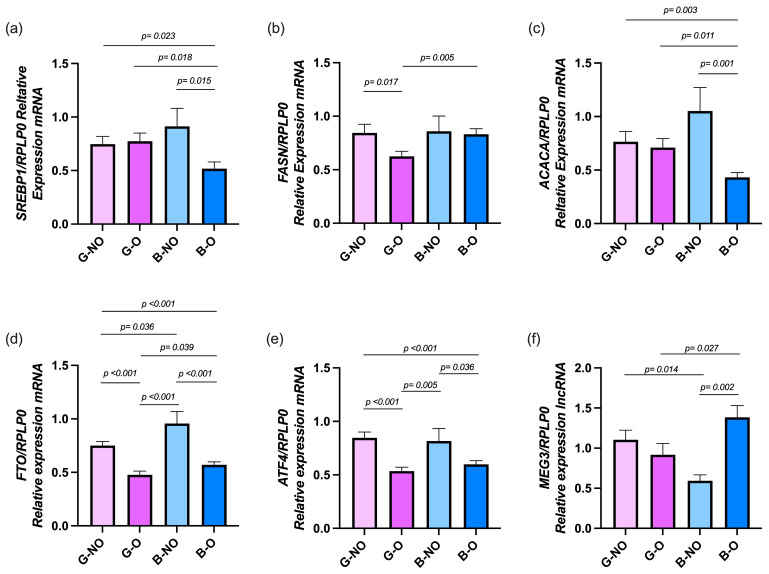
The expression of lipogenic genes in girls and boys with and without obesity. Gene expression of (**a**) *SREBP1*, (**b**) *FASN*, (**c**) *ACACA*, (**d**) *FTO*, (**e**) *ATF4*, and (**f**) *MEG3*. mRNA expression was measured by RT-qPCR using *RPLP0* as the reference gene and calculated using the 2^−ΔΔCt^ method. Statistical differences were evaluated by two-way ANOVA followed by Tukey’s post hoc test. Results are presented as mean ± SD. G-NO: Girls non-Obesity; G-O: Girls Obesity; B-NO: Boys non-Obesity; B-O: Boys Obesity.

**Figure 3 nutrients-17-02546-f003:**
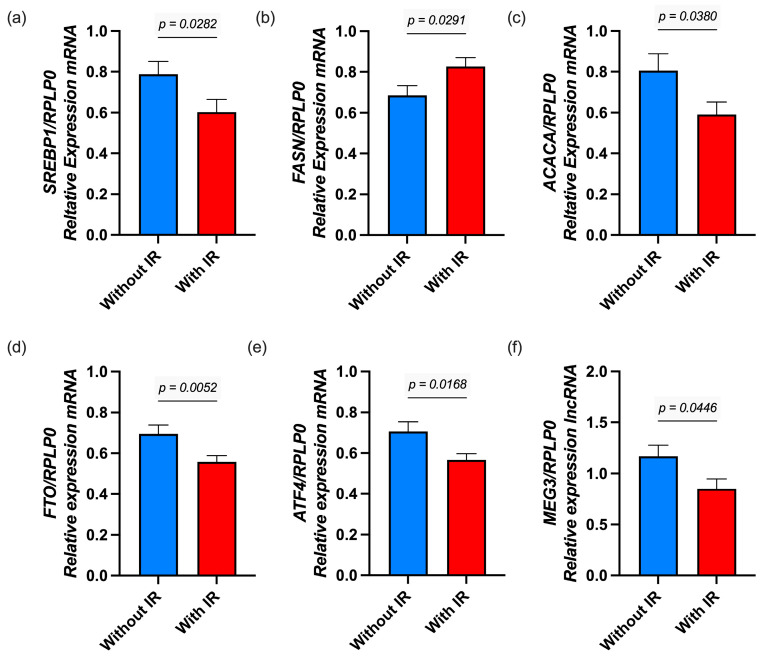
The expression of lipogenic genes in children with and without IR. Gene expression of (**a**) *SREBP1*, (**b**) *FASN*, (**c**) *ACACA*, (**d**) *FTO*, and (**e**) *ATF4*, and (**f**) *MEG3*. mRNA expression was measured by RT-qPCR using RPLP0 as the reference gene and calculated using the 2^−ΔΔCt^ method. Statistical differences were analyzed using the Mann–Whitney U test. Results are presented as mean ± SD.

**Table 1 nutrients-17-02546-t001:** Demographic, clinical, and metabolic parameters.

Variable	Total (*n* = 71)	Without Obesity (*n* = 27)	With Obesity (*n* = 44)	*p*-Value
Age (y)	8.94 ± 1.680	9.111 ± 1.740	8.84 ± 1.660	0.514 ^a^
Sex, *n* (%)				
Male	28 (39.44)	9 (32.14)	19 (67.86)	0.410 ^b^
Female	43 (60.56)	18 (41.86)	25 (58.14)	
Waist circumference (cm)	74.338 ± 13.462	60.833 ± 7.302	82.625 ± 8.831	<0.001 ^a^
Hip circumference (cm)	82.047 ± 12.037	70.940 ± 6.962	88.863 ± 9.031	<0.001 ^a^
BMI (kg/cm^2^)	22.725 (16.405–25.184)	16.113 (15.527–16.645)	24.292 (23.787–25.410)	<0.001 ^c^
BMI percentile	76.352 ± 30.283	41.555 ± 20.905	97.704 ± 1.373	<0.001 ^a^
Glucose (mg/dL)	87.288 ± 7.991	88.185 ± 6.314	86.738 ± 8.890	0.463 ^a^
Insulin (µIU/mL)	14.430 (8.400–21.400)	8.400 (7.200–9.550)	17.880 (15.000–22.100)	<0.001 ^c^
HOMA-IR	3.066 (1.767–4.796)	1.794 (1.596–2.092)	3.726 (3.123–4.801)	<0.001 ^c^
≥3, *n* (%)	36 (50.70)	6 (22.22)	30 (68.18)	<0.001 ^b^
Total CHO (mg/dL)	164.253 ± 27.204	164.185 ± 31.027	164.295 ± 24.953	0.986 ^a^
HDL-C (mg/dL)	47.983 ± 10.577	52.981 ± 8.716	44.915 ± 10.529	<0.001 ^a^
LDL-C (mg/dL)	85.982 ± 25.146	86.937 ± 28.916	85.396 ± 22.866	0.804 ^a^
TG (mg/dL)	129.7 (89.100–219.300)	100.600 (85.345–137.136)	157.050 (123.967–200.893)	0.009 ^c^
Added sugar (g)	73.724 ± 36.195	71.751 ± 44.585	74.934 ± 30.710	0.766 ^a^
≥50 g, *n* (%)	58 (81.69)	20 (74.07)	38 (86.36)	0.194 ^b^
Total energy intake (Kcal)	2383.5 (1983–2787)	2340 (1908–2698)	2490 (2085–2870)	0.294 ^c^

BMI: Body mass index; HOMA-IR: Homeostatic model assessment of insulin resistance; CHO: Cholesterol; HDL-C: High-density lipoprotein cholesterol; LDL-C: Low-density lipoprotein cholesterol; TG: Triglycerides. Results are expressed as mean ± standard deviation (SD) for parametric data or as median (interquartile range) for non-parametric data. ^a^ Student’s *t* test; ^b^ Chi-square test; ^c^ Mann–Whitney test.

**Table 2 nutrients-17-02546-t002:** Quantile regression of the expression of *MEG3* and its association with expression of genes involved in lipogenesis, adjusted for sex, age, and obesity.

Gene	Adjusted for Sex, Age, and Obesity	
	β	*p*-Value
*SREBP1*	0.322	<0.001
*FASN*	0.178	0.002
*ACACA*	0.180	0.030
*FTO*	0.146	0.004
*ATF4*	0.149	0.001

*SREBP1*: Sterol Regulatory Element-Binding Protein 1; *FASN*: Fatty Acid Synthase; *ACACA*: Acetyl-CoA Carboxylase; *FTO*: Fat Mass and Obesity-Associated Gene; *ATF4*: Activating Transcription Factor 4. β values correspond to unstandardized regression coefficients obtained from quantile regression models. *p* Values < 0.05 were considered statistically significant.

**Table 3 nutrients-17-02546-t003:** Quantile regression analysis of gene expression and its association with clinical and metabolic parameters, adjusted for sex, age, and obesity.

Variable	*SREPB1*		*FASN*		*ACACA*		*FTO*		*ATF4*		*MEG3*	
	β	*p*-Value	β	*p*-Value	β	*p*-Value	β	*p*-Value	β	*p*-Value	β	*p*-Value
Waist circumference (cm)	−0.012	0.266	−0.000	0.895	−0.006	0.572	0.000	0.873	0.000	0.873	0.000	0.981
Hip circumference (cm)	0.003	0.804	−0.000	0.958	−0.003	0.713	0.000	0.926	0.000	0.926	−0.005	0.830
Glucose (mg/dL)	−0.020	0.026	0.007	0.211	−0.011	0.099	−0.007	0.079	−0.007	0.079	0.010	0.576
Insulin (µIU/mL)	−0.012	0.030	0.005	0.123	−0.005	0.270	−0.004	0.148	−0.004	0.148	−0.008	0.415
HOMA-IR	−0.050	0.049	0.025	0.089	−0.026	0.234	−0.017	0.146	−0.017	0.146	−0.041	0.429
Total CHO (mg/dL)	−0.003	0.241	−0.001	0.474	0.001	0.575	−0.000	0.727	−0.000	0.727	0.001	0.674
HDL-C (mg/dL)	0.001	0.880	−0.008	0.037	0.006	0.397	0.002	0.443	0.002	0.443	0.007	0.592
LDL-C (mg/dL)	−0.002	0.418	−0.001	0.519	0.002	0.345	−0.000	0.485	−0.000	0.485	0.000	0.969
TG (mg/dL)	−0.000	0.652	0.001	0.052	−0.001	0.152	−0.000	0.753	−0.000	0.753	0.001	0.475

HOMA-IR: Homeostatic Model Assessment of Insulin Resistance; CHO: Cholesterol; HDL-C: High-Density Lipoprotein Cholesterol; LDL-C: Low-Density Lipoprotein Cholesterol; TG: triglycerides; *SREBP1*: Sterol Regulatory Element-Binding Protein 1; *FASN*: Fatty Acid Synthase; *ACACA*: Acetyl-CoA Carboxylase Apha; *FTO*: Fat Mass and Obesity-Associated Gene; *ATF4*: Activating Transcription Factor 4; *MEG3*: Maternally Expressed Gene 3. β values correspond to unstandardized regression coefficients obtained from quantile regression models. *p* Values < 0.05 were considered statistically significant.

**Table 4 nutrients-17-02546-t004:** Quantile regression analysis of the added sugar intake and its association with expression of genes involved in lipogenesis, adjusted for sex and total energy intake.

Gene	Adjusted for Sex and Total Energy Intake	
	β	*p*-Value
*SREBP1*	−0.360	0.050
*FASN*	−0.024	0.809
*ACACA*	−0.211	0.111
*FTO*	−0.221	0.032
*ATF4*	−0.050	0.426
*MEG3*	−0.272	0.396

*SREBP1*: Sterol Regulatory Element-Binding Protein 1; *FASN*: Fatty Acid Synthase; *ACACA*: Acetyl-CoA Carboxylase Alpha; *FTO*: Fat Mass and Obesity-Associated Gene; *ATF4*: Activating Transcription Factor 4; *MEG3*: Maternally Expressed Gene 3. β values correspond to unstandardized regression coefficients obtained from quantile regression models. *p* Values < 0.05 were considered statistically significant.

## Data Availability

The raw data supporting the conclusions of this article will be made available by the authors on request due to ethical restrictions and the need to protect participant confidentiality in accordance with institutional guidelines.

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
