# Peer review of "Altered Expression of the MEG3, FTO, ATF4, and Lipogenic Genes in PBMCs from Children with Obesity and Its Associations with Added Sugar Intake"

_nutrients, 2025, doi:10.3390/nu17152546_

Round 1

Reviewer 1 Report

Comments and Suggestions for Authors

< !--StartFragment -->

Dear Authors,

Thank you for the opportunity to read the results and review this article. The research problem is highly significant for the public health of most of the world's population. The increasing prevalence of excess body weight in the pediatric population is gaining momentum. Any efforts by scientists to thoroughly understand the mechanisms of this phenomenon are highly desirable. Gene expression studies, very modern, offer us all hope for perhaps identifying the sources of the global obesity pandemic.

In my opinion, a drawback of this study and a challenge for the authors is the wide range of chronological ages of the children studied, as well as the two genders, combined with the small sample size. These children certainly have very different developmental (biological) ages.

Boys and girls differ significantly in body fat content even before puberty, and after puberty, and even during puberty (and girls, especially obese ones, are certainly already menstruating), they experience a significant increase in body fat. Therefore, during puberty, hormonal and lipid profiles change spontaneously. All analyses should include gender, chronological age, and level of sexual maturity as determinant variables. Correlation analyses between a gene and, for example, added sugar are flawed if confounding variables are not taken into account. Certainly, the mechanisms of action of the analyzed genes may differ between the sexes and depend on developmental age. The authors should employ multivariate analyses that will identify real relationships. Appropriate references to these issues must also be made in the theoretical chapters.

Minor comments:

-Keywords should be different from those in the title.

-The description of anthropometric measurements should be more precise.

-Result interpretations, as in line 194, should be in the discussion, not the results.

-Table entries should be standardized (e.g., p-value).

-Figures are difficult to read; tabular comparisons are preferable.

Kind regards

reviewer

< !--EndFragment -->

Author Response

Reviewer: Thank you very much for your valuable and insightful comments on the protocol. Your observations are greatly appreciated and will contribute to improving the quality of the study.

1. All analyses should include gender, chronological age, and level of sexual maturity as

determinant variables.

R= We truly appreciate your valuable observation regarding pubertal development, We

fully agree that pubertal status, such as assessed by Tanner staging, could provide a more

accurate interpretation of metabolic parameters. However, in the context of our study, it

was not feasible to include clinical assessment of Tanner stage for several reasons. First,

this evaluation requires the examination of secondary sexual characteristics, which

involves collecting highly sensitive information from participants and their families.

Based on our previous experience, requesting such information can lead to discomfort,

lower willingness to participate, and even higher dropout rates—especially in children

with obesity, where social stigma is already a concern.

To address developmental differences, we chose to stratify analyses by sex, age and BMI

which we believe is a reasonable alternative within the context of our study.

2. Multivariate analysis

R= Thank you for your comments. In response to your suggestions, in addition to the

correlation analysis, we performed multivariate association analyses adjusted for

potential confounding variables, including sex, age, and BMI as shown in

Supplementaries Tables. These adjustments were made to account for potential

confounding factors and to identify meaningful associations.

3. Appropriate references to these issues must also be made in the theoretical chapters.

R= Thank you for your valuable commentaries. According with your suggestions, we

added the following text in the Introduction section:

“In addition, biological factors such as age and sex influence gene expression and

metabolic responses. During puberty, hormonal changes—such as increased estrogen or

testosterone—can modify fat distribution [16], insulin sensitivity [17,18], and lipid levels

[17], contributing to sex- and age-specific metabolic profiles. Several studies have

reported that the expression of metabolic genes, including FTO [19] and SREBP1 [20],

varies by sex.” (Page 2; Lines 80 – 84).

Moreover, in Discussion section, we included the following text:

“Although our analyses were adjusted for sex and age, the lack of pubertal assessment

(e.g., Tanner staging) may limit the interpretation of some findings, particularly those

related to metabolic and gene expression variability. Future studies should consider

incorporating pubertal status to better characterize developmental influences on gene

regulation.” (Page 14 – 15; Lines 494 – 498).

Finally, in Study Limitations section, we added the following statement:

“Another limitation is that pubertal status was not assessed using Tanner staging. While

this could have provided a more precise understanding of biological variability and

metabolic changes, its implementation would have required clinical evaluations and

handling of sen-sitive information, which could have reduced participation.” (Page 15;

Line 506 – 510).

4. Keywords should be different from those in the title

R= Thank you for your observations. We have change the Keywords from “LncRNA

MEG3; Pediatric Obesity; Lipid Metabolism, Added Sugar Intake” to “Gene regulation,

Pediatric Population, Non-coding RNA, Dietary sugar consumption”.

5. The description of anthropometric measurements should be more precise.

R= Thank you for your commentaries. In accordance with your suggestions, we have

added a new subsection titled “Anthropometric Measurements,” where we explain the

details as recommended. (Page 3; Lines 119 – 146)

6. Result interpretations, as in line 194, should be in the discussion, not the results.

R= Thank you for your observations. In accordance with your suggestions, we have

removed interpretations of results from the Results section.

7. Table entries should be standardized (e.g., p-value).

R= Thank you for your comments. Following your suggestion, we have standardized the

nomenclature for p-values throughout the tables.

8. Figures are difficult to read; tabular comparisons are preferable.

R= Thank you for your comment. In response to your suggestion, we reformatted the

figures to improve their clarity and readability.

Reviewer 2 Report

Comments and Suggestions for Authors

In the manuscript nutrients-3776434, the authors demonstrated MEG3–FTO–ATF4 axis and lipid metabolism-related gene expression are altered in children with obesity. This study has strength in measuring multiple variables, including clinical parameters and added sugar consumption, and determining the association between the variables; however, there are some concerns and issues required to be addressed.

  1. Please write full names of the variables in table 1. (e.g. waist circ -> waist circumference).
  2. To compare effects of sex and obesity on lipid metabolism-related gene expression, it will be better to combine figure 2 and 3. (for comparison among boys with obesity, boys without obesity, girl with obesity, and girls without obesity)
  3. Since figure 4 shows effects of insulin resistance, the authors should show the number and percentage of the participants with insulin resistance in Table 1.
  4. One of the most important results of this study is a negative correlation between added sugar intake and lipid metabolism-related gene expression. It will be better to change table 4 to scatter plots to see the correlation.

Author Response

Reviewer:  Thank you very much for your valuable and insightful comments on the manuscript. Your observations are greatly appreciated and will contribute to improving the quality of the study.

1. Please write full names of the variables in table 1. (e.g. waist circ -> waist

circumference).

R= Thank you for your observation. In accord your suggestion, the names of the

variables were full written in the table 1.

2. To compare effects of sex and obesity on lipid metabolism-related gene expression,

it will be better to combine figure 2 and 3. (for comparison among boys with

obesity, boys without obesity, girl with obesity, and girls without obesity)

R = Thank you for your commentaries. Following your suggestion, we combined

Figures 2 and 3 to allow for comparison among all groups. Additionally, we

modified the Results section to reflect the changes in the new figure (Page 6; Lines

229–244).

Corresponding adjustments were also made in the Discussion section to incorporate

the new findings (Page 12; Lines 377 – 383; Lines 407 – 411; and Page 13; Lines

456 – 459) .

3. Since figure 4 shows effects of insulin resistance, the authors should show the

number and percentage of the participants with insulin resistance in Table 1.

R= Thank you for your observation. We confirm that the number and percentage of

children with insulin resistance were already included in Table 1, specifically in the

Total column under the variable HOMA-IR ≥ 3, n (%). This cutoff point (HOMA-IR

≥ 3) has been used to define insulin resistance in Mexican children, and corresponds

to 36 participants (50.70%).

4. One of the most important results of this study is a negative correlation between

added sugar intake and lipid metabolism-related gene expression. It will be better to

change table 4 to scatter plots to see the correlation.

R= Thank you for your suggestions. According to your commentaries, we replaced

the data from Table 4 with scatter plots in Figure 4. As suggested, this figure includes

both the Rho and P-value.

Reviewer 3 Report

Comments and Suggestions for Authors

Multi Digital Publishing Institute (MDPI): mdpi-nutrients-3730192-v1

Article

Title: Altered Expression of the MEG3–FTO–ATF4 Axis and Lipid Metabolism Genes in PBMCs from Children with Obesity and Its Associations and Added Sugar Intake

The authors performed a study to assess gene expression of certain genes in PBMCs and lipid levels in children with obesity.

Overall comments:

Overall, the authors’ thought processes of choosing these genes for analysis are unclear.

Genes (not their protein products) should be italicized.

I see that there are some manuscripts published on what the authors call MEG3-FTO-ATF4 axis, but this really does not make sense as far as genetic and biological pathways are concerned.  Please see below for details about each gene.

When one talks about some type of axis, there should be definitive connection between regions or genes, such as hypothalamus-pituitary-adrenal (HPA) axis.

It is important that each gene function is considered in detail rather than choosing randomly.

For obesity, there are genes which are causal genes for monogenic obesity as well as associated genes so their expression would be more relevant if the authors are trying to understand obesity.

Mahmoud R, Kimonis V, Butler MG. Genetics of Obesity in Humans: A Clinical Review. Int J Mol Sci. 2022 Sep 20;23(19):11005. doi: 10.3390/ijms231911005. PMID: 36232301; PMCID: PMC9569701.

For LDL metabolism, there are other genes which are specific for their pathways such as LDLR, APOB, PCSK9, etc.  There are separate genes involved in TG metabolism.

Thus, if the authors are trying to involve genes or gene expressions for a study, through understanding of each genes as well as their protein product (if they exist) is imperative.  MEG3 is not translated and it has other special properties as noted below.

Please consider the following.

Biochemical pathways and lipoproteins pathways are interconnected somewhere overall, but there are so many steps involved.  Thus, the results from this study are not totally well-thought (they seem very random) through.  Of course, there are many transcription changes that occur with any type of stress or dietary changes, but there has to be some underlying rational reasoning for choosing these gene changes, and from the manuscript, it is uncertain.

Definitely, triglyceride metabolism is closely connected to fatty acid synthesis because a molecule of triglyceride is made up of one glycerol (backbone) and three fatty acids attached, but TG metabolism itself in circulation is totally different.  TG molecules are carried in chylomicrons and very-low-density lipoproteins.  Chylomicrons are in the exogenous pathway in which triglycerides in diet are contained. 

Glucose pathway is the pathway which everyone studied in biology or chemistry as the first lesson. 

It is always important to think about biological/biochemical/genetic pathways in this type of study; otherwise, it does not make sense and the results really do not represent what the authors are trying to explain.

Specific comments:

Introduction

Lines 56-59:

Long non-coding RNA (lncRNAs) …

This sentence is not accurate and misleading.  They are non-coding which means that they do not code for proteins.  Please learn about lncRNAs.

Long noncoding RNAs (lncRNAs) are a type of RNA molecules that are typically longer than 200 nucleotides.  They may be involved in a wide range of cellular processes, including gene expression regulation, chromatin modification, and RNA splicing. 

LncRNAs may act as molecular scaffolds, decoys, guides, and signals, influencing gene expression at various levels. 

Yao RW, Wang Y, Chen LL. Cellular functions of long noncoding RNAs. Nat Cell Biol. 2019 May;21(5):542-551. doi: 10.1038/s41556-019-0311-8. Epub 2019 May 2. PMID: 31048766.

Tsai MC, Manor O, Wan Y, Mosammaparast N, Wang JK, Lan F, Shi Y, Segal E, Chang HY. Long noncoding RNA as modular scaffold of histone modification complexes. Science. 2010 Aug 6;329(5992):689-93. doi: 10.1126/science.1192002. Epub 2010 Jul 8. PMID: 20616235; PMCID: PMC2967777.

It is true that lncRNAs may play a role in the development of IR, but there are so many steps which are involved biologically so it is not straightforward as the authors wrote.

MEG3 is an imprinted gene found in the imprinted region of chromosome 14q32.2 which is only expressed on the maternal allele and the paternal MEG3 is silenced by methylation.

It has been shown to play a significant role in human development and is implicated in various biological processes, including cancer, brain development, and cellular senescence. 

Except for SREBP1c and FASN, the others are not specifically lipid metabolism genes.

It is important that the authors understand different types of genes which are causal genes for monogenic disorders vs genes found through GWAS.  These are totally different, and they should not be treated in the same way.

ATF4, or Activating Transcription Factor 4, is a protein that acts as a transcription factor, meaning it regulates target gene expression.

As the name suggests, the protein ATF4 is an activating transcription factor, and it regulates its gene targets (for transcription).   It plays a role in the cellular response to various stresses, including cellular stress and oxidative stress.

ATF4 is involved in regulating cell survival, metabolism, and apoptosis, and it has been implicated in various diseases, including cancer. 

FTO gene, also known as the fat mass and obesity- “ASSOCIATED” gene, is a gene identified through GWAS associated with obesity.  This is totally different from a causal gene for disorders. 

FTO (FTO alpha-ketoglutarate-dependent dioxygenase)

Biallelic pathogenic variants in FTO have been found in children with growth retardation, developmental delay, facial dysmorphism in a consanguineous family.

It primarily functions as an alpha-ketoglutarate-dependent dioxygenase, specifically acts as a RNA-demethylase that removes methyl groups from RNA.  This process impacts gene expression and protein synthesis.

 Boissel S, Reish O, Proulx K, Kawagoe-Takaki H, Sedgwick B, Yeo GS, Meyre D, Golzio C, Molinari F, Kadhom N, Etchevers HC, Saudek V, Farooqi IS, Froguel P, Lindahl T, O'Rahilly S, Munnich A, Colleaux L. Loss-of-function mutation in the dioxygenase-encoding FTO gene causes severe growth retardation and multiple malformations. Am J Hum Genet. 2009 Jul;85(1):106-11. doi: 10.1016/j.ajhg.2009.06.002. Epub 2009 Jun 25. PMID: 19559399; PMCID: PMC2706958.

FASN (fatty acid synthase)

It encodes for an enzyme fatty acid synthase which catalyzes the conversion of acetyl-CoA and malonyl-CoA, in the presence of NADPH, into long-chain saturated fatty acids.

Jensen-Urstad AP, Semenkovich CF. Fatty acid synthase and liver triglyceride metabolism: housekeeper or messenger? Biochim Biophys Acta. 2012 May;1821(5):747-53. doi: 10.1016/j.bbalip.2011.09.017. Epub 2011 Oct 8. PMID: 22009142; PMCID: PMC3288544.

ACACA (Acetyl-CoA Carboxylase-alpha)

Acetyl-CoA carboxylase is essential for fatty acid synthesis.

Biallelic pathogenic variants cause acetyl-CoA carboxylase deficiency in which individuals are unable to properly produce fatty acids.  Individuals have a range of symptoms such as hypotonia, developmental delays, muscle weakness and seizures in some. 

Acetyl-CoA carboxylase catalyzes formation of the key precursor that fuels FASII, the conversion of acetyl-CoA to malonyl-CoA. Acetyl-CoA is a key intermediate in many pathways

Lines 73-74:

Please clarify this sentence.

Which lipid are the authors referring as “lipid biosynthesis”?

Lipoprotein metabolism is pretty well deciphered, and this sentence does not make sense.

Do they mean cholesterol synthesis, triglyceride synthesis, vs fatty acid synthesis.  Their metabolic pathways are totally separate as well as their regulation.

Cholesterol synthetic pathway and lipoprotein pathway are both deciphered, and they are totally different.

Duan Y, Gong K, Xu S, Zhang F, Meng X, Han J. Regulation of cholesterol homeostasis in health and diseases: from mechanisms to targeted therapeutics. Signal Transduct Target Ther. 2022 Aug 2;7(1):265. doi: 10.1038/s41392-022-01125-5. PMID: 35918332; PMCID: PMC9344793.

Cholesterol and triglyceride are carried in various lipoproteins, and it is important that the authors understand these differences.

When authors are analyzing lipoproteins such as LDL-C (low-density lipoprotein cholesterol), and HDL-C (high-density lipoprotein cholesterol), cholesterol contained in these lipoproteins are measured.  On the other hand, triglycerides are either in very-low density lipoproteins or in chylomicrons.  HDL pathway is totally different from the others.

VLDL -> IDL -> LDL

Glucose metabolism is also totally different.

Fatty acid synthesis is another different pathway.

Please be aware of biological and biochemical pathways when performing a study involving genetics.

There are many other genes which are directly responsible for lipoprotein synthesis.

It is important that a study should be based on biological processes and biochemical pathways. 

With obesity and metabolic syndrome, TG levels are often increased, and HDL-C levels are decreased due to these lipoprotein interactions of their pathways.

It is important that monogenic and GWAS genes should be handled differently.

Monogenic genes have well identified functions, but GWAS genes, especially SNPs may or may not have any function.

Until these biological and biochemical pathways are thought through, any result from this type of study is not credible.

Thank you very much for allowing me to review this manuscript. 

Sincerely,

Comments on the Quality of English Language

Sentences are fine, but it is important that the sentences align with biological processes.  

Author Response

Reviewer:  Thank you very much for your valuable and insightful comments on the manuscript. Your observations are greatly appreciated and will contribute to improving the quality of the study.

1. I see that there are some manuscripts published on what the authors call MEG3-

FTO-ATF4 axis, but this really does not make sense as far as genetic and

biological pathways are concerned. Please see below for details about each gene.

R= Thank you for your helpful observation. Following your observations, we

removed the term “axis” from the title and throughout the manuscript. We

understand that this term implies a well-defined biological or anatomical

pathway.

2. It is important that each gene function is considered in detail rather than choosing

randomly. For obesity, there are genes which are causal genes for monogenic

obesity as well as associated genes so their expression would be more relevant if

the authors are trying to understand obesity.

R= Thank you for your commentaries. In accord your suggestions, we revised the

introduction section to clarify the distinction between causal genes for monogenic

obesity and genes identified through GWAS. We explained that FTO is not a

causal gene but rather a susceptibility gene associated with obesity via regulatory

mechanisms (Page 3; Lines 91 – 95).

"Meanwhile, FTO, a gene identified through genome-wide association studies

(GWAS) as associated with obesity, plays a role in lipid accumulation by

enhancing the maturation and nuclear translocation of SREBP1c [26]. Unlike

causal genes for monogenic obesity, such as MC4R or LEP, which have a direct

role in early-onset and severe forms of obesity, FTO influences susceptibility

through regulatory mechanisms [27]."

In accord to ATF4, we have revised the description of this gene in the introduction

section to clarify that although ATF4 is not a classical lipid metabolism gene

(Page 2 – 3; Lines 87 – 90).

"For instance, ATF4 is a transcription factor involved in cellular stress responses.

Although not a classical lipid metabolism gene, it contributes to lipogenesis [24].

Its deficiency has been associated with reduced hepatic lipogenesis through

downregulation of SREBP1c, PPARγ, and FASN expression [25]."

3. For LDL metabolism, there are other genes which are specific for their pathways

such as LDLR, APOB, PCSK9, etc. There are separate genes involved in TG

metabolism.

R= Thank you for your commentaries. In response, we have addressed this point

in the limitations section by explicitly acknowledging that our study did not

include key genes involved in lipoprotein metabolism, such as LDLR or APOB

(Page 15; Lines 511 – 513).

"Moreover, this study focused on genes related to lipogenesis and the metabolic

stress response, but did not include key genes involved in lipoprotein metabolism,

such as LDLR or APOB, which limits the scope of metabolic pathway analysis.

4. Lines 56–59: The sentence on long non-coding RNAs (lncRNAs) is inaccurate

and misleading. lncRNAs are non-coding, meaning they do not code for proteins.

They may be involved in a wide range of cellular processes, including gene

expression regulation, chromatin modification, and RNA splicing. Please revise

accordingly."

R= Thank you for your valuable observation. In accordance with your suggestion,

we revised the description of lncRNAs in the Introduction section and included a

more detailed explanation about their functions (Page 2; Lines 58–63).

"Long non-coding RNAs (lncRNAs), which are RNA molecules longer than 200

nucleotides, do not encode proteins but play important regulatory roles at the

transcrip-tional, post-transcriptional, and epigenetic levels [5]. They can act as

molecular scaffolds, guides, decoys, or signals, influencing gene expression,

chromatin remodeling, and RNA splicing [6]. LncRNAs are involved in several

biological processes, such as the regulation of lipid and glucose metabolism and

inflammatory pathways—hallmarks of IR and obesity [7]."

We believe this updated version is now more accurate and clearly reflects the

biological roles of lncRNAs.

5. Lines 73-74: Please clarify this sentence. Which lipid are the authors referring as

“lipid biosynthesis”? Lipoprotein metabolism is pretty well deciphered, and this

sentence does not make sense. Do they mean cholesterol synthesis, triglyceride

synthesis, vs fatty acid synthesis? Their metabolic pathways are totally separate

as well as their regulation.

R= Thank you for your commentaries. In response, we have replaced the term

“lipid biosynthesis” with “lipogenesis” to accurately reflect the intracellular

synthesis of fatty acids and triglycerides, which is the process we refer to

throughout the manuscript.

Round 2

Reviewer 1 Report

Comments and Suggestions for Authors

Dear Authors,
Thank you for the opportunity to review the manuscript again and for the authors' responses.
I understand the authors' position regarding the difficulties in obtaining information on the age of sexual maturity. However, internationally validated questionnaires for its assessment should not pose a barrier to obtaining this data. 

I thank you for using multivariate statistical methods that take into account age, gender, and BMI. Personally, I believe that the results presented in the Supplementary Materials are more valuable than the results of the basic correlations available in the main text. Only the use of analyses that take into account confounding variables, such as age, gender, and BMI, into account, yields reliable results worth discussing in the publication.

Kind regards,
reviewer

Author Response

  1. Thank you for the opportunity to review the manuscript again and for the authors' responses. I understand the authors' position regarding the difficulties in obtaining information on the age of sexual maturity. However, internationally validated questionnaires for its assessment should not pose a barrier to obtaining this data. 

R= We really apreciate your observation. We agree that validated cuestionaries can be very useful to asses pubertal stage, but in our study it was not posible to apply them because of ethical restrictions and we didn’t had the specific consent to colect that type of sensitive information. We have added this as a limitation in the discussion.

  1. Personally, I believe that the results presented in the Supplementary Materials are more valuable than the results of the basic correlations available in the main text. Only the use of analyses that take into account confounding variables, such as age, gender, and BMI, yields reliable results worth discussing in the publication.

R= Thank you for your observation. Based on this recomendation, the multivariate analysis ajusted for sex, age and obesity status were moved from the Supplementary Material to the main Results and are now shown as the principal findings. The simple correlations are left only as exploratory data in the Supplementary Tables. In this pediatric cohort, the participants were divided into two clinical groups, normal weight and obesity, so the models were ajusted for obesity status (normal weight vs. obesity) instead of using BMI as a continuous variable, since this reflects better the clinical classification of adiposity in our study design.